# Why Concurrent CDDP and Radiotherapy Has Synergistic Antitumor Effects: A Review of In Vitro Experimental and Clinical-Based Studies

**DOI:** 10.3390/ijms22063140

**Published:** 2021-03-19

**Authors:** Shinsuke Nagasawa, Junko Takahashi, Gen Suzuki, Yamazaki Hideya, Kei Yamada

**Affiliations:** 1Department of Radiology, Graduate School of Medical Science, Kyoto Prefectural University of Medicine, Kyoto 602-8566, Japan; gensuzu@koto.kpu-m.ac.jp (G.S.); yamahi@koto.kpu-m.ac.jp (Y.H.); kyamada@koto.kpu-m.ac.jp (K.Y.); 2Biomedical Research Institute, National Institute of Advanced Industrial Science and Technology (AIST), Ibaraki 305-8566, Japan; junko-takahashi@aist.go.jp

**Keywords:** cis-diamminedichloroplatinum (cisplatin: CDDP), radiotherapy (RT), chemo-radiotherapy (CRT), concurrent, synergistic effect, radio-sensitizing

## Abstract

Chemo-radiotherapy, which combines chemotherapy with radiotherapy, has been clinically practiced since the 1970s, and various anticancer drugs have been shown to have a synergistic effect when used in combination with radiotherapy. In particular, cisplatin (CDDP), which is often the cornerstone of multi-drug combination cancer therapies, is highly versatile and frequently used in combination with radiotherapy for the treatment of many cancers. Therefore, the mechanisms underlying the synergistic effect of CDDP and radiotherapy have been widely investigated, although no definitive conclusions have been reached. We present a review of the combined use of CDDP and radiotherapy, including the latest findings, and propose a mechanism that could explain their synergistic effects. Our hypothesis involves the concepts of overlap and complementation. “Overlap” refers to the overlapping reactions of CDDP and radiation-induced excessive oxidative loading, which lead to accumulating damage to cell components, mostly within the cytoplasm. “Complementation” refers to the complementary functions of CDDP and radiation that lead to DNA damage, primarily in the nucleus. In fact, the two concepts are inseparable, but conceptualizing them separately will help us understand the mechanism underlying the synergism between radiation therapy and other anticancer drugs, and help us to design future radiosensitizers.

## 1. Introduction

Chemo-radiotherapy (CRT), which involves the administration of anticancer drugs and radiotherapy (RT), has been performed since the 1970s. The radio-sensitizing effect of cis-diamminedichloroplatinum (cisplatin: CDDP) was first reported by Zak et al. in 1971 [1].

Early CRTs were used to treat patients for whom no alternatives remained, such as patients with cancers in the head and neck area, and the results were significantly inferior to RT alone or surgery. In part, this was because radiation in the 1970s was an extension of simple X-ray photography, so the accuracy of tumor irradiation was low, and thus the adverse events often exceeded the therapeutic effects.

In the 1980s, a three-dimensional RT planning device was developed and the accuracy of irradiation to tumors improved, along with the results of CRT. Beginning in the 1990s, a series of reports showed that the overall survival by CRT far exceeded that by RT alone in patients with head and neck cancer [2,3,4,5], and in 2009 a study showed that there was no difference in overall survival between surgery and CRT for stage I esophageal cancer [6,7].

The reason for adopting CRT to control cancer is that while chemotherapy is a systemic therapy, and thus complements the local antitumor effects of RT, it also has radio-sensitizing effects which enhance the effects of RT. Naturally, however, CRT has more significant adverse events than RT alone.

The antitumor effects of CDDP are highly versatile, such that CDDP-based CRTs are applied to a wide variety of cancers, including head & neck, esophageal, gastric, lung, urothelial, cervical, ovarian, testicular, skin, hematologic, and osteosarcoma cancers. A great deal of research has been dedicated to exploring the mechanisms underlying the synergism between CDDP and RT, but no definitive conclusions have been reached. One of the mechanisms proposed to date is that the combination of CDDP and radiation enhances the DNA damage induced by either monotherapy, but recent studies have suggested the existence of other mechanisms as well [8,9,10,11].

In this report, we overview findings on the radio-sensitization mechanism of CDDP from both cell biological studies and clinical trials, and then propose a new concept which may explain the synergistic mechanism between CDDP and RT: the main mechanism of this synergistic effect is the overlapping of excessive oxidation reactions by CDDP and RT, respectively.

Please note that, here, RT refers to RT by X-ray. Heavy ion beam, boron-neutron capture, and proton beam therapies are not included in this review, because their physical and biological reactions are different from those of X-ray RT. In addition, for assessing the additive/synergistic effects of CRTs in general, the isobologram [12] and combination Index [13] have been proposed in in vitro studies.

## 2. Proposed Mechanisms of the Synergistic Effects of CDDP and RT

It is generally thought that CDDP becomes cytotoxic by covalently binding to DNA to form interstrand and intrastrand crosslinks, and thereby inhibiting DNA replication and RNA transcription [14,15,16,17].

When CDDP is used in combination with radiation, there are a few different proposed mechanisms for radio-sensitization, and they are as follows: (1) the free radicals generated by ionizing radiation increase in the presence of the transition element (Pt) [16,17]; (2) free electrons released from irradiated DNA are captured by Platinum agent, and the DNA is chemically damaged [16,18]; (3) Platinum agent induces G2 arrest in the cell cycle which is complemented by high radiosensitivity in the G2 phase [16,17]; (4) Platinum agent uptake is enhanced by irradiation [16,17,19]; (5) Platinum agent is radio-sensitizer for hypoxic cell (hypoxic cell is usually radio-resistant) [17,18,20]; and (6) Platinum agent inhibits the repair of radiation-generated single-strand breaks (SSBs) and double-strand breaks (DSBs) in DNA [16,17,18,19,20]. However, there has been no definitive evidence of the predominance of any one of these mechanisms, and thus the question has remained unanswered from the 1970s to the present.

## 3. Studies Contradicting the Proposed Mechanisms of Action of CDDP

There are two issues that contradict with the abovementioned classical view of cytotoxicity.

First, some previous reports have suggested that, contrary to the proposed mechanisms listed above, there is actually little interaction between CDDP and nuclear DNA [21,22,23]. Assuming that the quantity of DNA in one cell is about 6.5 pg, as calculated from Avogadro’s number, the weight of CDDP bound to DNA is reported to range from 10^−3^ to less than 10^−2^ compared to that of intracellular CDDP (Table 1). Thus, it can be concluded that the quantity of CDDP bound to DNA is substantially lower than that of intracellular CDDP.

These facts contradict the main, conventional explanations of the cytotoxicity of CDDP, i.e., that it forms interstrand and intrastrand crosslinks with DNA. They evoke the question of whether a very small quantity of CDDP bound to DNA would have a critical effect on cytotoxicity.

Second, the combined effect of CDDP and radiation is not an additive effect, but a synergistic effect [16,17,24,25]. This synergistic effect is maximized by administering CDDP within 6 h before or after RT, and if the gap exceeds 24 h, a synergistic effect will not be obtained by the combination [25,26]. This fact that the synergistic effect appears not only in radiation after CDDP administration but also in radiation before CDDP administration, in addition to the above-mentioned reports that the quantity of CDDP bound to DNA is very small, suggest that the synergistic effect cannot be explained merely by the mechanism of CDDP bound to DNA.

Note that this time interval (1 to 6 h) for the synergistic effect coincides with the time for the cellular metabolic reaction to CDDP [8,27], the time for maintaining the crosslink between CDDP and DNA [28,29,30], and the time for DNA repair after radiation [31,32,33].

## 4. Synergistic Mechanism of CDDP and RT

As stated above, most of the administered CDDP is DNA-unbound, so we hypothesized that the antitumor effect of CDDP is exerted mainly in the cytoplasm, not in the nucleus. Additionally, from the reports that the cytotoxicity of CDDP depends on the degree of oxidative loading caused by CDDP [8,10,11] and that the oxidative loading by radiation determines the cytotoxicity [9], we considered that the main mechanism of the cytotoxicity of CDDP and radiation may be excessive oxidative loading in the cytoplasm. In addition, it has been reported that irradiation of the cytoplasm rather than the nucleus strongly induces cell death in studies using a ferroptosis inducer [34]. Therefore, we considered that the reaction in the cytoplasm by CDDP and radiation could reasonably be referred to as the major reaction. Additionally, the conventional reactions in which interacting CDDP binds to DNA and in which radiation cleaves nuclear DNA could be considered the minor reactions.

In a major reaction, excessive oxidative load triggers biological responses having the form of a cascade. In particular, the excessive oxidative loading on the cells caused by radiation is also inherited by the progeny cells of the directly irradiated cells, and further, cells distant from the directly irradiated cells also undergo excessive oxidative load-like changes (i.e., a bystander effect: cell damages outside the irradiated field being similar to those inside the irradiated field due to intercellular communication via gap-junctions and inflammatory cytokines [35,36,37]). From these facts, the changes in cells due to excessive oxidative loading can be considered to be caused by metabolism. These metabolic reactions are biological processes in which malfunctions gradually accumulate over time, and are on a different order of time than chemical processes in which DNA is simply cleaved by radiation. These facts are consistent with the fact that, in clinical experience, tumor shrinkage after radiotherapy (including concurrent CDDP) generally occurs within several months.

## 5. Major Reaction of the Radio-Sensitization of CDDP: Overlapping of the Same Reaction

Cytotoxicity due to radiation and CDDP is mediated by a common mechanism of excessive oxidative loading. Excessive oxidative loading (Figure 1A) indirectly disrupts intracellular signaling by reducing intracellular reduction equivalents, resulting in the loss of normal cell function, including DNA repair; (Figure 1B) causes mitochondrial dysfunction (such as malfunctions in the citric acid cycle and electron transfer system); and (Figure 1C) causes oxidative damage to the lipids, proteins, and nucleic acids through direct chemical reactions [8,9,10,18,28].

CDDP reduces the intracellular NADH/NAD^+^ and NADPH/NADP^+^ ratios rapidly (within 1–6 h) in a dose-dependent manner. These ratios recover quickly within one day. Glycolysis in tumor cells is susceptible to fluctuation of intracellular reducing equivalents. The conversion of pyruvate to lactic acid, which requires NADH, is particularly vulnerable [8,10,11,38]. Decreased reducing equivalents result in the disruption of redox homeostasis and metabolic impairment of the Warburg effect. This leads to the death of tumor cells [8].

In many types of tumor cells, a decrease in the intracellular lactate level (in correlation with a tumor growth inhibitory effect) is induced immediately after administration of CDDP at a micromolar concentration that has been shown to induce an antitumor effect. This indicates that CDDP exerts a wide range of antitumor effects [8]. In addition, the primary target of excessive oxidative loading caused by CDDP is the mitochondria, and mitochondrial dysfunction due to oxidative loading causes cell death [28] and generates reactive oxygen species (ROS) derived from mitochondrial metabolism [9]. ROS also induce extrinsic/intrinsic apoptosis pathways [11,28].

Excessive oxidative loading directly causes damage to cell constituents such as nucleic acids, proteins and lipids. In particular, regarding lipid peroxidation, it has been reported that not only organelle dysfunction is associated with peroxidation, but also antitumor effects are brought about by the induction of ferroptosis with concomitant depletion of intracellular reducing equivalents such as Glutathione (GSH) [34].

On the other hand, radiation also causes excessive oxidative loading within cells. ROS generated by radiation temporarily (for about 2 h) reduces the NADH/NAD^+^ ratio [38]. This means that, like CDDP, radiation triggers the perturbation of redox homeostasis. Furthermore, when ionizing radiation is applied, ·OH is generated by water decomposition. Radiation also stimulates induction of intracellular nitric oxide synthase (NOS) activity to produce a large quantity of NO. In the nucleus, ·OH reacts with DNA to cause DNA damage, including DNA strand breaks. In an oxygen-rich environment, ·O_2_^−^, the strongest oxidative agent, is generated from ·OH [9].

In mitochondria, excessive oxidative loading causes mitochondrial dysfunction, and electrons leaked from the mitochondria react with O_2_ to generate ·O_2_^−^. ·O_2_^−^ generated by radiation in mitochondria or in an oxygen-rich environment reacts with organic substances to generate ROS such as RO_2_^−^ or reacts with NO to generate RNS such as ONOO^−^, and increases the oxidative damage of organelles. In addition, mitochondrial dysfunction caused by ionizing radiation continues to impose a sustained oxidative loading on cells [9].

It has also been reported that neutrophils are involved in ROS production after radiation. Additionally, in neutrophils, ROS production increases. At 24 h after radiation, the number of neutrophils in the tumor area reaches a maximum and then gradually decreases. When the neutrophils are experimentally extinguished, the antitumor effect of RT is attenuated [39]. From this fact, it is highly possible that neutrophils induced after radiation produce ROS, and that the ROS in turn contribute to the antitumor effect.

As described above, the mechanisms underlying the biochemical antitumor effects of RT and CDDP are homologous in terms of excessive oxidative loading. Each of these homologous mechanisms lowers the threshold of the other antitumor effects, and as a result, a synergistic effect is obtained when RT and concurrent CDDP are used (Figure 1). In support of the hypothesis that RT and CDDP have homologous antitumor effects, there is the clinically common experience that tumors resistant to CDDP are also resistant to RT.

If this radio-sensitization mechanism of CDDP is correct, a drug having an antitumor effect mechanism homologous to that of RT will have a synergistic effect when used in concurrent combination with RT. It was reported that doxorubicin may have an antitumor effect similar to that of CDDP [26]. Doxorubicin is also a drug having a wide range of antitumor effects, and is mainly used in hematological malignancies, breast cancer, ovarian cancer, and bone and soft tissue tumors. However, doxorubicin has cardiotoxicity, and this toxicity is enhanced when it is used concurrently with RT. Hence, the use of them to the chest is problematic.

## 6. Why Do Duplicates of the Same Reaction Result in a Synergistic Effect?

We considered the mechanisms that cause synergistic effects rather than additive effects by overlapping the same reactions, such as RT and CDDP (Figure 2). Systems connected in series do not exhibit significant dysfunction in response to minor failure (Figure 2: ②), but the more the failures increase, the more exponentially their function deteriorates (Figure 2: ③). Therefore, in metabolic cascade reactions from ROS generation to cell damage, it is presumed that the threshold for cell damage drops sharply when CDDP and RT, which have the same mechanism of cell damage, are used concurrently. This may be the reason why a synergistic effect appears instead of an additive effect when CDDP and RT are used together.

This hypothesis is supported by the finding that CDDP and RT exhibit a synergistic effect even when CDDP is administered at a concentration below the cytotoxic concentration [40,41]. In other words, even if they are not cytotoxic on their own, the fact that they produce synergistic rather than additive effects when combined corroborates that their reactions have the same mechanism.

## 7. Minor Reactions of Radio-Sensitization of CDDP: CDDP and RT Complement Each Other’s Functions

It is generally considered that minor reactions may have a mechanism in which CDDP inhibits the repair of SSBs produced by RT and promotes SSB formation (Figure 2) [8,16,17,20]. Since the crosslink of DNA by CDDP is metabolized within a few hours, it may be necessary to perform RT within a few hours after the administration of CDDP to obtain a sufficient synergistic effect [27,42,43,44].

In addition to the rapid metabolism of the CDDP reaction, because CDDP-induced DNA damage is randomly distributed on the DNA, while radiation-induced DNA damage occurs in clusters locally in the DNA [45,46,47], it can be inferred that a large bolus administration of CDDP will be necessary to obtain sufficient synergism between the antitumor effects of CDDP and RT. This hypothesis is consistent with the results of clinical research [48]. In addition, it was reported that platinum agents bind to the SH or S-S groups of radical scavengers such as glutathione and DNA repair enzymes such as Poly (ADP-ribose) polymerase (PARP), and then alters their conformation to regulate their activity [18,28]. A mechanism was also proposed in which platinum agents modify the indirect action of ionizing radiation to increase the production of radiation-induced SSBs [37]. (Figure 3).

Note that the biochemical aspects of the minor reactions of RT and CDDP are different until they result in DNA damages. On the other hand, those of the major reaction of RT and CDDP are almost the same, throughout the accumulation of oxidative load. In this respect, the major reaction and the minor reactions are decisively different.

## 8. Possibility of a Diminished Antitumor Effect by the Combined Use of CDDP and RT

There are also reports showing that radiation (especially low doses of low LET radiation) and CDDP induce resistance to each other. The combined use may enhance the DNA repair ability and induce adaptive responses including antioxidant reactions [8,9,25].

The detailed mechanism of why low doses of CDDP or radiation work in favor of cell survival has not been elucidated. This response may be a phenotype of the organism’s universal defense capability against poisons, X-ray, and other disorders.

With regard to RT, this is well known as radiation hormesis [49,50]: low-dose irradiation not only induces radiation tolerance, but may even have positive physical effects (i.e., Radon hot-springs). A similar phenomenon to the hormesis effect is the ischemic preconditioning (IPC) observed in the brain and heart [51,52,53,54]. This is a response in which a minor ischemic attack leads to a better outcome of subsequent severe ischemic symptoms. Similar phenomena may be empirically observed in other organs.

In addition, superoxide radicals generated by CDDP or radiation can act as both initiators and terminators of free radical-mediated chain reactions that result in lipid peroxidation, etc. [9]. Depending on the timing of CDDP administration, CDDP may offset the effects of RT.

## 9. Consistency between Clinical Trials and the Mechanism of RT Combined with CDDP

In this section, we will consider the results of three clinical trials of RT combined with CDDP for head and neck cancer [48,55,56]. We believe the results of these trials support, or at least do not contradict, the mechanism of action of CDDP and RT that we proposed above.

The protocol regimens of each arm of the clinical trials are shown in Figure 4.

### 9.1. Clinical Trial 1

Clinical trial 1 [55] is a prospective study evaluating the effect of chemotherapy in addition to RT for unresectable head and neck carcinoma. The patients were randomized in each arm, and approximately 100 patients were collected per arm.

In clinical trial 1, CCRT outperformed the RT alone arm in major outcomes such as local control and overall survival, indicating that RT with concurrent CDDP has a high antitumor effect (Table 2). RT alone protocol is often used for patients for whom CDDP cannot be administered due to renal/hepatic impairment or old age.

### 9.2. Clinical Trial 2

Clinical trial 2 [56] is a prospective study to evaluate the endpoints of chemotherapy and RT for Stage 3 & 4 resectable larynx carcinoma. Patients were randomized in each arm, and approximately 170 patients were recruited. The patients were randomized in each arm and approximately 170 patients were collected per arm.

As shown in clinical trial 2, the sequential RT arm performed better in the major outcomes than the RT alone arm, and the concurrent RT arm performed even better than the sequential RT arm. From this result, it can be seen that concurrent administration is necessary in order to obtain the maximum synergistic antitumor effect in RT combined with CDDP (Table 3). The sequential administration may have achieved only the additive effect of chemotherapy alone and RT alone.

Of note is that there was no significant difference in distant metastatic control between the two groups, so even if the CDDP and RT are concurrent, they are effective only for local control.

Sequential RT is given to patients who have had a good response to prior chemotherapy to achieve radical cure of cancer. Sequential RT or chemotherapy may also be given to patients who cannot tolerate the adverse effects of CCRT.

### 9.3. Clinical Trial 3

Clinical trial 3 [48] is a prospective study to evaluate the endpoints of CCRT with every 3 weeks CDDP and CCRT with once a week low-dose CDDP for locally advanced head & neck carcinoma. The patients were randomized in each arm and approximately 150 patients were collected per arm.

Clinical trial 3 shows that the antitumor effect of CDDP is not duration time dependent, but dose dependent (Table 4). The higher the dose of CDDP per administration, the higher the antitumor effect when concurrently combined with radiation, and the administration of CDDP every 3 weeks is basically performed with the clinical regimen.

Why does low-dose CDDP have a poor antitumor effect? It can be inferred that while high-dose CDDP immediately subjects tumors to excessive oxidative loading, some damages caused by low-dose CDDP will be resolved over time without the tumors receiving sufficient oxidative loading to provide antitumor effects [25,26]. This is also the case with DNA damage, because high-dose CDDP can cause DNA damage more widely, to complement the cluster-like DNA damage due to radiation.

CCRT of the once-a-week CDDP regimen is common in CCRT of the pelvic region, such as for cervical cancer, because of the low incidence of adverse events.

## 10. Future Prospects and Considerations

CDDP is a highly versatile anticancer drug that exerts an antitumor effect on many types of tumors. Therefore, we anticipate that the two basic concepts of the radio-sensitization mechanism of CDDP—namely, (1) overlapping of the same reaction and (2) complementation of each other’s functions—can be applied to other anticancer drugs. In addition, these two concepts may be useful for predicting the degree of radio-sensitization or for understanding the mechanism of anticancer drugs when combined with RT. However, the actual intracellular reactions would represent an admixture of these two concepts, and thus in practical terms the concepts are probably inseparable. It is not hard to imagine that excessive oxidative loading causes many DNA damages.

RT is generally performed in fractionated doses (typically, 2 Gy/fraction), because it can reduce damage to normal tissues compared to single high dose. The question is whether that can be fully explained by the hypotheses proposed herein. It is empirically known that radiation has a large effect on cells that are actively dividing. Given that fact, we can speculate that normal cells are less affected by excessive oxidative loading because they have an inactive metabolism related to cell division, which means that normal cells suffer only limited and resolvable damage from fractionated irradiation.

## 11. Conclusions

We outlined the mechanisms of radio-sensitization of CDDP with the latest findings. We speculate that the mechanism of radio-sensitization by CDDP involves not only the conventional reactions in the nucleus, but also reactions in the cytoplasm, including the mitochondria, which greatly contribute to the radio-sensitization.

The mechanism of action in the nucleus is that RT and CDDP complement each other’s unique properties related to DNA cleavage. On the other hand, the reaction mechanism in the cytoplasm, which is considered to be the major reaction, involves the overlapping and accumulation of excessive oxidative loading by radiation and CDDP.

## Figures and Tables

**Figure 1 ijms-22-03140-f001:**
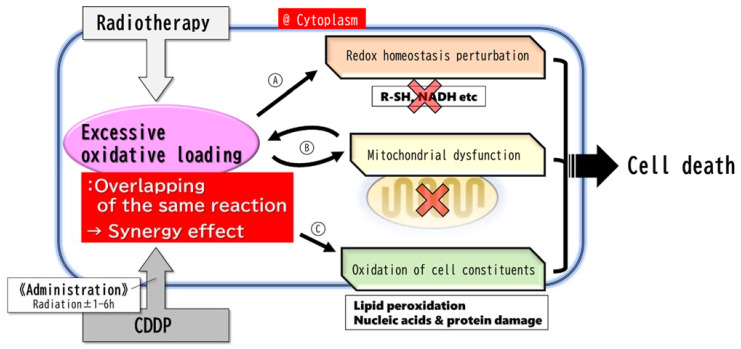
Major reaction. The major reaction occurs mainly in the cytoplasm. Radiation and CDDP give cells an identical excessive oxidative loading response, resulting in a synergistic effect leading to cell death. The metabolic response to the excessive oxidative loading by CDDP reaches its maximum within 6 h and is rapidly resolved thereafter. Therefore, it is necessary to administer CDDP within 6 h before or after radiation in order to obtain the synergistic effect. (**A**) Radical scavengers and NADH/NAD+ rate decrease → redox homeostasis perturbation → metabolic (Warburg effect, DNA repair, etc.) malfunctions. (**B**) Mitochondrial dysfunctions → more excessive and persistent oxidative stress, and apoptosis. (**C**) Oxidation of cell constituents (especially lipid peroxidation) → cell dysfunctions, destruction of cell components (DNA, etc.), ferroptosis.

**Figure 2 ijms-22-03140-f002:**
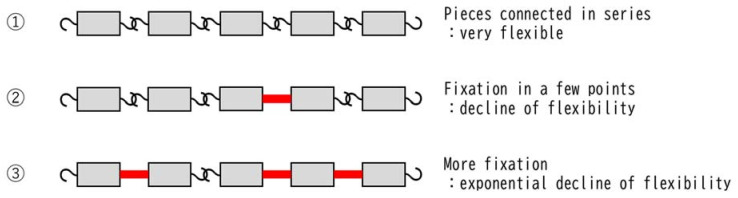
Mechanism underlying the synergistic effect produced by overlapping of the same reaction. Chains lose their flexibility exponentially each time their joints are fixed. The same can be said for the cascade of normal metabolic reactions, which are intracellular chain reactions. It is well-known that cells can maintain homeostasis even if they sustain some damage, but homeostasis suddenly collapses when a certain amount of damage is accumulated. The schema in this figure illustrates that ① unfixed chains are very flexible and can take many forms; ② the chain retains some flexibility even if the joint is fixed in several places; and ③ as the number of fixed joints increases, the forms that the chain can take decrease exponentially and are extremely limited.

**Figure 3 ijms-22-03140-f003:**
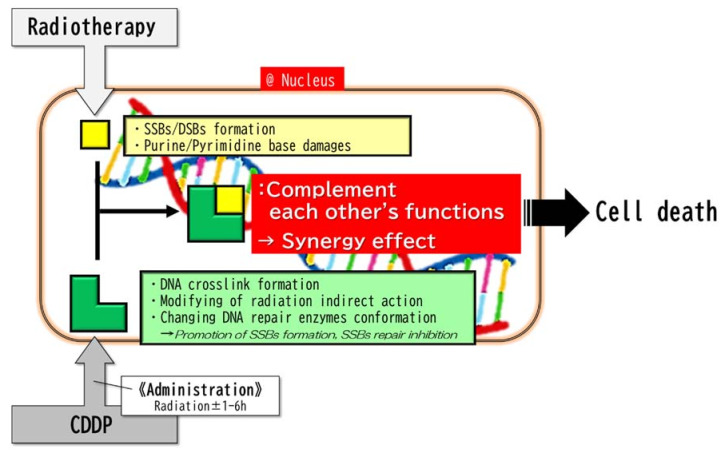
Minor reactions. The minor reactions take place in the nucleus. The complementary combination of radiation-induced primary DNA strand breaks and CDDP-induced DNA damages (DNA crosslink, etc.) synergistically results in DNA damage and then cell death. Since the crosslinks of DNA by CDDP are metabolized in a few hours, it may be necessary to perform RT within a few hours after the administration of CDDP in order to obtain a sufficient synergistic effect. Abbreviations: SSBs, single-strand breaks; DBSs, double-strand breaks.

**Figure 4 ijms-22-03140-f004:**
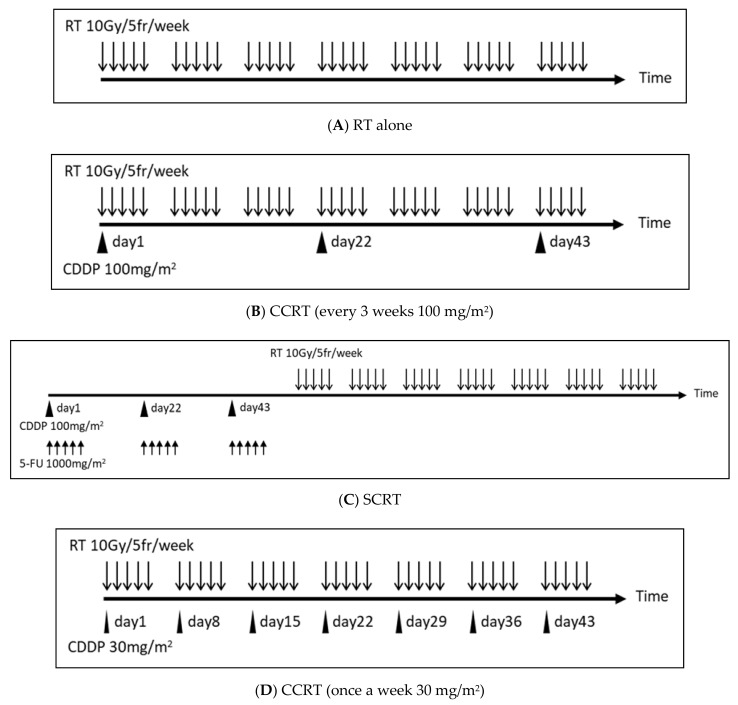
The protocol regimens of each arm of three clinical trials of RT plus CDDP for head and neck cancer. (**A**) Do 2 Gy per 1 fr, 5 times a week. Continue this for 7 weeks. (**B**) Administer CDDP every 3 weeks (day 1, day 22, day 43) within a few hours before or after RT. The CDDP administration is 100 mg/m^2^ per dose. (**C**) Initially, administer 5-FU for 5 consecutive days + simultaneous CDDP every 3 weeks (the addition of 5-FU increases the intensity of chemotherapy). Perform RT sequentially. The CDDP dose is 100 mg/m^2^ per dose, the 5-FU dose is 1000 mg/m^2^ per dose. (**D**) Administer a small quantity of CDDP once a week within a few hours before or after RT. The CDDP dose is 30 mg/m^2^ per dose. Abbreviations: RT, radiotherapy; fr, fraction; CCRT, concurrent chemo-radiotherapy; SCRT, sequential chemo-radiotherapy; Gy, gray; mg, milligram; m^2^, square meter.

**Table 1 ijms-22-03140-t001:** Ratio of intracellular CDDP and DNA-bound CDDP.

Cell Line	Administered CDDP [μM]	Time for Measure CDDP Quantity	DNA-Bound CDDP /Intracellular CDDP (Approximate Value)	Reference
HCT116	7.5	4 h	1/500	Tippayamontri, 2011 [21]
HeLa	10.0	4 h	1/650	Nikolić, 2016 [22]
PC9	NR	NR	1/1000	Imai, 2016 [23]

The ratio of DNA-bound CDDP is extremely small compared to that of intracellular CDDP. Abbreviations: μM, micro mol/dm^3^; NR, not reported.

**Table 2 ijms-22-03140-t002:** Clinical trial 1 RT vs. CCRT.

	LC [%]	3 Years OS [%]	MS [Month]	3 Years DSS [%]
RT	27.4	23.0	12.6	33.0
CCRT (100 mg/m^2^)	40.2	37.0	19.1	51.0

Results showing a synergistic effect when CDDP and RT are combined. The CCRT arm has better results, including with respect to the LC and OS, compared to the RT alone arm. Abbreviations: RT, radiotherapy; CCRT, concurrent chemo-radiotherapy; LC, local control; OS, overall survival; MS, median survival; DSS, disease-specific survival.

**Table 3 ijms-22-03140-t003:** Clinical trial 2 RT vs. SCRT vs. CCRT.

	5 Years LRC [%]	10 Years LRC [%]	5 Years DFS[%]	10 Years DFS [%]	5 Years DC[%]	10 Years DC[%]
RT	53.6	50.1	28.0	14.8	78.0	76.0
SCRT	58.2	53.7	37.7	20.4	85.3	83.4
CCRT (100 mg/m^2^)	71.1	69.2	38.0	21.6	86.4	83.9

Results showing that concurrent CDDP is required for a maximum antitumor effect. The sequential arm performed better than the RT alone arm in major outcomes such as LRC, but the concurrent arm performed even better. Abbreviations: RT, radiotherapy; SCRT, sequential chemo-radiotherapy; CCRT, concurrent chemo-radiotherapy; LRC, locoregional control; DFS, disease-free survival; DC, distance control; mg, milligram; m^2^, square meter.

**Table 4 ijms-22-03140-t004:** Clinical trial 3 CCRT (once a week 30 mg/m^2^) vs. CCRT (every 3 weeks 100 mg/m^2^).

	2 Years LRC [%]	mPFS [Month]	mOS [Month]	Acute G3–4 [%]
CCRT (once a week 30 mg/m^2^)	58.5	17.7	~40	71.6
CCRT (every 3 weeks 100 mg/m^2^)	73.1	28.6	~40	84.6

As a characteristic of CDDP, the results show that a bolus administration of high dose CDDP is desirable for the maximum antitumor effect. It can be inferred that higher CDDP doses result in a temporary, extensive and excessive oxidative loading reaction or DNA damage (see Figure 2). G3–4 are adverse event grades that require hospitalization. Abbreviations: CCRT, concurrent chemo-radiotherapy; PFS, progression-free survival; OS, overall survival; mg, milligram; m^2^, square meter.

## Data Availability

Not applicable.

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
