# Peer review of "Why Concurrent CDDP and Radiotherapy Has Synergistic Antitumor Effects: A Review of In Vitro Experimental and Clinical-Based Studies"

_ijms, 2021, doi:10.3390/ijms22063140_

Round 1
Reviewer 1 Report
General comments:
The authors overview findings on the radio-sensitization mechanism of CDDP from both cell biological studies and clinical trials, and then propose a concept which may explain the synergistic mechanism between CDDP and RT.
Major comments:
- line 59: “then propose a new concept which may explain the synergistic mechanism between CDDP and RT”. What kind of new concept should be briefly described here.
- line 220-222: “This hypothesis is supported by the finding that CDDP and RT exhibit a synergistic effect even when CDDP is administered at a concentration below the cytotoxic concentration [40,41].”. I cannot understand why these three lines can support this effect. Please add more detailed explanation or description at the end of this paragraph.
- line 283: “The protocol regimens of each arm of the clinical trials are shown in Figure 4.” Although the authors provided the figure, I suggest the authors to provide several sentences for a brief description for these figure content. Finally, the authors should make comments for these protocol in the end of this section.
- line 279: These three clinical trials should be briefly listed here with short name or description.
- References were cited in number but listing in a alphabet order. Please use suitable reference formatting.
- Most of the references were old. If possible, please use the update references if available. Since the review may provide a comprehensive view. The new information is encouraged to cite more.
Minor comments:
- line 65: references for [13 cis34] need to correct.
- line 78: The references [8,16,17,18,19,20] should be individually cites after (1) to (6).
Author Response
Thank you very much for your sincere review. We are thankful for the time and energy you expended.
Based on your comments, we have made some corrections. We have attached the version of the corrected paper and would appreciate if you could refer to it.
Major comments:
- line 59: “then propose a new concept which may explain the synergistic mechanism between CDDP and RT”. What kind of new concept should be briefly described here.
(Added the following) The main mechanism of the synergistic effect is the overlapping of excessive oxidation reactions by CDDP and RT, respectively.
- line 220-222: “This hypothesis is supported by the finding that CDDP and RT exhibit a synergistic effect even when CDDP is administered at a concentration below the cytotoxic concentration [40,41].”. I cannot understand why these three lines can support this effect. Please add more detailed explanation or description at the end of this paragraph. 
(Added the following) In other words, even if they are not cytotoxic on their own, the fact that they produce synergistic rather than additive effects when combined corroborates that their reactions have the same mechanism.
- line 283: “The protocol regimens of each arm of the clinical trials are shown in Figure 4.” Although the authors provided the figure, I suggest the authors to provide several sentences for a brief description for these figure content. Finally, the authors should make comments for these protocol in the end of this section. 
In Figure 4, we summarize each protocol (A~D).
(At the end of the clinical trial1 section, we added the following) RT alone protocol is often used for patients who cannot use CDDP due to renal/hepatic impairment or old age.
(At the end of the clinical trial1 section, we added the following) Sequential RT is given to patients who have had a good response to prior chemotherapy to achieve radical cure of cancer. Sequential RT or chemotherapy may also be given to patients who cannot tolerate the adverse effects of concurrent chemoradiotherapy.
We have included comments on the clinical trial 3 protocol at the end of clinical trial 3 section.
- line 279: These three clinical trials should be briefly listed here with short name or description.
(At the start of the clinical trial1 section, we added the following) Clinical trial 1 [55] is a prospective study evaluating the effect of chemotherapy in addition to RT for unresectable head and neck carcinoma. The patients were randomized in each arm, and approximately 100 patients were collected per arm.
(At the start of the clinical trial2 section, we added the following) Clinical trial 2 [56] is a prospective study to evaluate the endpoints of chemotherapy and RT for Stage 3 & 4 resectable larynx carcinoma. Patients were randomized in each arm, and approximately 170 patients were recruited. The patients were randomized in each arm and approximately 170 patients were collected per arm.
(At the start of the clinical trial3 section, we added the following) Clinical trial 3 [48] is a prospective study to evaluate the endpoints of CCRT with every 3 weeks CDDP and CCRT with once a week low-dose CDDP for locally advanced head & neck carcinoma. The patients were randomized in each arm and approximately 150 patients were collected per arm.
- References were cited in number but listing in a alphabet order. Please use suitable reference formatting.
I'm sorry, there may be some mistake. References are listed by number.
- Most of the references were old. If possible, please use the update references if available. Since the review may provide a comprehensive view. The new information is encouraged to cite more.
We are also concerned about the issue you pointed out. However, since our review includes conventional hypothesis and no literature analogous to our new hypothesis has been published as far as we can tell, there will inevitably be fewer references to new hypothesis than to conventional one.
Minor comments:
- line 65: references for [13 cis34] need to correct.
We've fixed it.
- line 78: The references [8,16,17,18,19,20] should be individually cites after (1) to (6).
We've fixed it.

Reviewer 2 Report
Excellent work. The manuscript is nicely written and presented. It should be accepted in the present form.
Author Response
Thank you very much for your kindful review. We are thankful for the time and energy you expended.
Based on the comments of other referee, we have made some corrections. We have attached the version of the corrected paper and would appreciate if you could refer to it.

Round 2
Reviewer 1 Report
All reviewer's concerns have been well responded.